# Traces of electron-phonon coupling in one-dimensional cuprates

Ta Tang [1,2], Brian Moritz[2], Cheng Peng[2], Zhi-Xun Shen [1,2,3,4] & Thomas P. Devereaux [2,4,5] ✉

The appearance of certain spectral features in one-dimensional (1D) cuprate materials has been attributed to a strong, extended attractive coupling between electrons. Here, using time-dependent density matrix renormalization group methods on a Hubbard-extended Holstein model, we show that extended electron-phonon ($e$–$ph$) coupling presents an obvious choice to produce such an attractive interaction that reproduces the observed spectral features and doping dependence seen in angle-resolved photoemission experiments: diminished $3k_F$ spectral weight, prominent spectral intensity of a holon-folding branch, and the correct holon band width. While extended $e$–$ph$ coupling does not qualitatively alter the ground state of the 1D system compared to the Hubbard model, it quantitatively enhances the long-range superconducting correlations and suppresses spin correlations. Such an extended $e$–$ph$ interaction may be an important missing ingredient in describing the physics of the structurally similar two-dimensional high-temperature superconducting layered cuprates, which may tip the balance between intertwined orders in favor of uniform $d$-wave superconductivity.

The origin of high-temperature superconductivity found in layered, quasi-two-dimensional (2D) cuprates remains a puzzle despite concerted, continuous investigations over the last few decades. From the perspective of numerical simulations, simplified models such as the Hubbard and $t$–$J$ Hamiltonians have been studied extensively, which have produced rich physics relevant to cuprates such as antiferromagnetism, stripes, and strange metal behavior[1–3]. However, evidence that these simplified models possess a uniform $d$-wave superconducting ground state remains elusive. Quasi-long-range superconductivity has only been reported on small-width cylinders[4–13], with strong competition from coexisting charge orders. Superconducting correlations decay exponentially on the hole doped side for wider clusters, indicating the superconductivity is absent for parameters thought to be relevant to hole-doped cuprates.

These findings indicate that the Hubbard model is incomplete, at least for describing the cuprates and high-temperature superconductivity. The inclusion of additional ingredients, such as phonons, which manifest as kinks or replica bands in photoemission measurements[14–18], may provide the crucial remedy. However, exact numerical simulations of the 2D Hubbard model are already challenging (the density matrix renormalization group (DMRG) method is limited by the growth of entanglement entropy and determinant quantum Monte Carlo (DQMC) and related methods suffer from the fermion sign problem); and adding bosonic degrees of freedom creates an even more daunting problem. The task may be made easier, with more numerical control, by turning to the simpler yet structurally similar, one-dimensional (1D) cuprates.

Recent angle-resolved photoemission spectroscopy (ARPES) experiments on the 1D cuprate $Ba_{2-x}Sr_xCuO_{3+\delta}$[19] provide an excellent platform for testing theoretical models. Modeling in 1D has both well-established theory, and numerical simulations that can be performed with a higher degree of control and accuracy. The measured single-

[1]Department of Applied Physics, Stanford University, California 94305, USA. [2]Stanford Institute for Materials and Energy Sciences, SLAC National Accelerator Laboratory, 2575 Sand Hill Road, Menlo Park, CA 94025, USA. [3]Department of Physics, Stanford University, Stanford, CA 94305, USA. [4]Geballe Laboratory for Advanced Materials, Stanford University, Stanford, CA 94305, USA. [5]Department of Materials Science and Engineering, Stanford University, Stanford, CA 94305, USA. ✉e-mail: tpd@stanford.edu

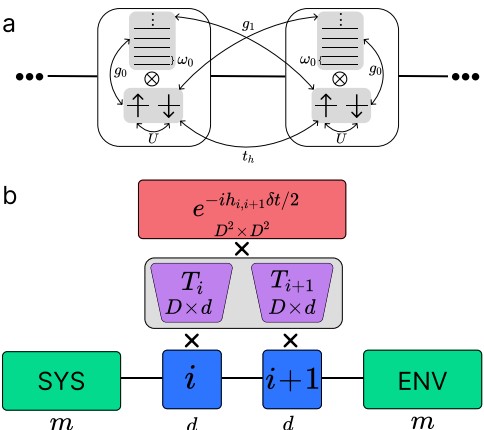

**Fig. 1 | Schematics for the model and dynamical LBO. a** Schematic for the one-dimensional Hubbard-extended Holstein model. On each site, the local Hilbert space is a direct product of phonon and charge degrees of freedom. The charges of opposite spin interact with an on-site repulsion $U$ and can hop to neighboring sites. Local phonons with a frequency $\omega_0$ couple to both on-site and nearest-neighbor charges. **b** Schematic for the dynamical LBO. We keep the dimension of the effective Hilbert space of the system and environment blocks as $m$, respectively. Each site $i$ has an optimized basis of dimension $d$. The wave function is transformed to a $D \times d$ bare basis ($D = D_{ch} \times D_{ph}$, where $D_{ch} = 4$ represents the local charge Hilbert space dimension, and $D_{ph}$ is the bare phonon basis dimension) through a $D \times d$ transformation matrix, *i.e.* $T_i$, before applying the time evolution gate of shape $D^2 \times D^2$. Subsequently, a new optimal basis and transformation $T_i$ are obtained; and the wave function is projected to the new optimal basis before moving on to the next gate.

particle spectra provide a detailed proving ground for assessing the impact of terms added to model Hamiltonians. ref. [19] showed that the simple Hubbard model fails to reproduce salient details of the spectra near the Fermi surface: a prominent holon-folding (hf)-branch emanates from $k_F$ and quickly fades away with doping. This spectral feature, and its doping dependence, can be well reproduced when one includes a strong nearest-neighbor attractive interaction $V \sim -t$ in the Hamiltonian. A natural near-neighbor attraction exists in the Hubbard model, evident when downfolding to the $t$–$J$ model, but such a weak attraction ($\sim -J/4$) cannot account for the observed effect. Rather, this strong attraction likely originates from extended electron–phonon ($e$–$ph$) coupling, as discussed in recent work[19,20].

To investigate the influence of the extended $e$–$ph$ coupling, in this paper, a time-dependent DMRG (tDMRG) method is employed to study the single-particle spectral function and ground state properties of a 1D Hubbard-extended Holstein model. The extended $e$–$ph$ coupling quantitatively reproduces the dominant hf-branch seen in experiments while also correctly reproducing the holon branch band width, matching the observed spectra. Approximating this model using an effective nearest-neighbor attraction V fails to reproduce all of these features. Moreover, while the extended $e$–$ph$ coupling does not qualitatively alter the ground state obtained from the Hubbard model, which qualitatively remains a Luttinger liquid with sub-dominant superconducting pair-field correlations that decay as a power law with distance, the results show that the extended $e$–$ph$ coupling quantitatively enhances the superconducting pair-field correlations by reducing the overall exponent, making them longer-ranged. It is surmised that in two dimensions, an extended $e$–$ph$ coupling may tip the balance between different phases and help to realize a dominant $d$-wave superconducting ground state.

## Results
### Models
To produce an effective nearest-neighbor attractive interaction between charges, we consider an optical phonon mode, which couples

to charge density beyond the local site. Previous estimates[20] have shown that this Hubbard-extended Holstein model can produce an effective interaction on par with that extracted from ARPES experiments[19] for a reasonable phonon frequency and $e$–$ph$ coupling strength. For simplicity and to achieve better numerical convergence, here, we consider only on-site and nearest-neighbor $e$–$ph$ coupling (see Fig. 1a). This Hubbard-extended Holstein Hamiltonian takes the form

$$H = H_{el} + \omega_0 \sum_i \hat{a}_i^\dagger \hat{a}_i$$
$$+ g_0 \sum_i \hat{n}_i(\hat{a}_i^\dagger + \hat{a}_i) + g_1 \sum_{\langle ij \rangle} \hat{n}_i(\hat{a}_j^\dagger + \hat{a}_j), \quad (1)$$

where $\hat{a}_i^\dagger$ and $\hat{a}_i$ are the phonon ladder operators on site $i$, $\hat{n}_i$ is the total charge number operator on site $i$, $\omega_0$ is the phonon frequency, $g_0$ is the on-site $e$–$ph$ coupling, $g_1$ is the nearest-neighbor $e$–$ph$ coupling, and $\langle ij \rangle$ sums over nearest-neighbors. $H_{el}$ denotes the electronic part of the Hamiltonian, a 1D single-band Hubbard model,

$$H_{el} = -t_h \sum_{\langle ij \rangle \sigma} (\hat{c}_{i\sigma}^\dagger \hat{c}_{j\sigma} + h.c.) + U \sum_i \hat{n}_{i\uparrow} \hat{n}_{i\downarrow}, \quad (2)$$

where $\hat{c}_{i\sigma}^\dagger$ ($\hat{c}_{i\sigma}$) is the charge creation (annihilation) operator on site $i$ for spin $\sigma$, $\hat{n}_{i\sigma}$ is the charge number operator on site $i$ for spin $\sigma$, and $U$ is the on-site repulsion. To avoid confusion with the time variable $t$, we use $t_h$ to denote the hopping integral. For comparison, we also evaluate the extended-Hubbard model, which introduces a nearest-neighbor attractive interaction,

$$H_v = H_{el} + V \sum_{\langle ij \rangle} \hat{n}_i \hat{n}_j, \quad (3)$$

where $\hat{n}_i$ and $\hat{n}_j$ are total charge number operators on neighboring sites.

Unless otherwise specified, we use the following parameters in our simulations: $U = 8t_h$, $\omega_0 = 0.2t_h$, $g_0 = 0.3t_h$, $g_1 = 0.15t_h$, and $V = -t_h$. The values chosen for $U$ and $V$ were those that produced the best fit of the ARPES experimental spectra using cluster perturbation theory (CPT)[21,22] for an effective extended-Hubbard model[19]; and the $e$–$ph$ couplings $g_0$ and $g_1$ fall within the range estimated in ref. [20]. Here, we use a larger phonon frequency than that used in ref. [20] for better numerical convergence, but expect that a smaller phonon frequency would produce a stronger effective attraction, which would further enhance the hf-branch; although, one would need to ensure that the stronger effective coupling would not lead to phase separation.

We use DMRG[23,24] to obtain the ground states of the models defined in Eqs. (1)–(3); and we use tDMRG[25–27] to obtain real-frequency spectra from the Fourier transform of time-dependent correlators of the form $\langle \hat{O}_i^\dagger(t)\hat{O}_j(0) \rangle$. To efficiently deal with the infinite phonon Hilbert space on each site, we adopt a local basis optimization (LBO) for the ground state[28] and a dynamical LBO for time evolution[29], as schematically shown in Fig. 1b. Details about the method and numerical simulation are provided in the Methods section.

### Single particle spectral function
Figure 2 displays the lesser Green's function $\mathcal{G}^<_{j,L/2,\uparrow}(t)$, defined as $\mathcal{G}^<_{mn\sigma}(t) = i\langle \hat{c}^\dagger_{m\sigma}(t)\hat{c}_{n\sigma}(0) \rangle$, and the corresponding single-particle spectra removal obtained for the Hubbard model on an 80-site chain at half-filling. In Fig. 2a, following the removal of an electron from the center of the chain, one can see that the propagator attains a significant value at the two chain ends within a time $T \sim 20t_h^{-1}$, which sets the maximum real-time propagation for the simulation. Padding the Green's function with zeros from time $T$ to time $2T$ limits the frequency resolution of a fast Fourier transform to $\omega_{n+1} - \omega_n = \pi/T \sim 0.16\,t_h$. This

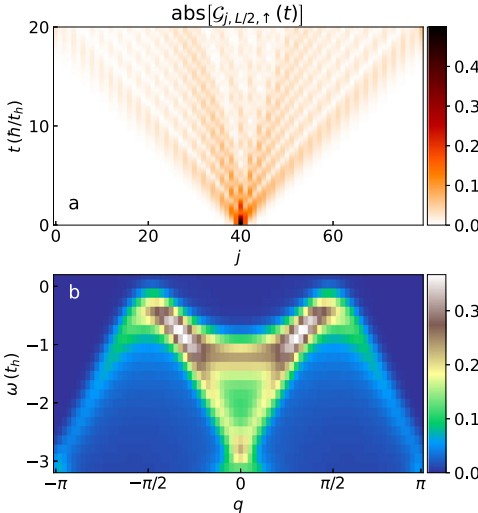

**Fig. 2 | Single-particle spectral function of Hubbard model at half-filling. a** The lesser Green's function $\mathcal{G}^<_{j,L/2,\uparrow}(t)$ for an 80-site chain at half-filling for the Hubbard model. Time is measured in units of $\hbar/t_h$ and $\hbar = 1$ in our calculation. We use a time step $\delta t = 0.04 t_h^{-1}$ and evolve the system for a total time $T = 20 t_h^{-1}$. **b** The single-particle spectral function obtained by Fourier transform of $\mathcal{G}^<$ in (**a**), with energy and momentum broadening of $\sigma_\omega = 0.2 t_h$ and $\sigma_k = 2\pi/L$, respectively.

provides a rather coarse resolution, but it is nevertheless more than adequate for comparison to the experimental ARPES spectra from the 1D chain cuprate, which is rather broad[19]. The single-particle spectrum, which is obtained using the tDMRG method and shown in Fig. 2b, agrees well with the results from cluster perturbation theory[19,21,22], dynamical DMRG, and the Bethe ansatz[30–33]. There are clear spinon and holon branches, demonstrating spin-charge separation in 1D. In the following, we use a chain of length $L = 80$ to compute and compare the single-particle spectral function of different models. A small broadening is used to give the spectra a high resolution, at least when compared with the experiment data, to better observe how different models affect the salient spectral features.

Figure 3a.1–6 shows the single-particle removal spectra of the Hubbard model across a range of doping. As observed in the experiment, splitting between the spinon and holon branches persists with doping. Our results correspond well to previous Hubbard model results on 1D and quasi-1D systems from dynamical DMRG and the Bethe ansatz[31,33], and also are consistent with spectra near the Fermi level from DQMC and DMRG calculations of the multi-band Hubbard model, which includes oxygen $p$-orbitals[34]. Here, we will focus on two spectral features: the branch of the removal spectrum emanating from $k_F$, which disperses downward toward $\pi$, hereafter the hf-branch, and the $3k_F$-branch (or more precisely $2\pi - 3k_F$), which also disperses downward toward $\pi$, but from $3k_F$. In the MDCs obtained from the Hubbard model (Fig. 3b.1–6), one sees that between these two features, the $3k_F$ peak is dominant. This result is contradictory to experimental observations, where the hf-peak is dominant, and the $3k_F$-peak is barely visible[19].

In Fig. 3c.1–6 and 3d.1–6, we confirm that adding a nearest-neighbor attractive interaction $V = -t_h$ enhances the hf-branch and produces spectra that are visibly more consistent with the experimental data at lower doping[19]. As we mentioned previously, this attractive interaction likely originates from $e$–$ph$ coupling. Here, we also simulate the underlying $e$–$ph$ Hamiltonian, with the results shown in Fig. 3e.1–6. Below 20% doping, one sees an enhanced hf-branch, while the $3k_F$-branch has been suppressed significantly by the $e$–$ph$ coupling (see Fig. 3e.1–3 and 3f.1–3). In all three models, the intensities in both the hf- and $3k_F$ branches become barely perceptible beyond ~20% doping. Using a larger broadening to compare more closely with

the experimental spectra and to extract intensities by fitting MDCs results in a doping-dependent intensity of hf-peak that matches well to the analyzed ARPES data (see Fig. S6 and Fig. S7 in Supplementary Information).

One significant difference between spectra for the extended Hubbard model and the Hubbard-extended Holstein model is that the nearest-neighbor attractive interaction in the extended Hubbard model significantly shrinks the holon bandwidth at higher doping (see Fig. 3c.1-6). In Fig. 4, we plot the holon binding energy at $k = 0$ as a function of doping to reflect the change in the holon bandwidth. By comparison, one sees that the results from the Hubbard-extended Holstein model are more consistent with the ARPES data, as the $e$–$ph$ interaction would renormalize the holon-branch only within ~$\omega_0$ of the Fermi energy.

## Ground state correlation functions

The good agreement with ARPES measurements begs the question: How does the extended $e$–$ph$ interaction affect the ground state? As a first step towards understanding this question, we study the ground state correlation functions of the 1D Hubbard-extended Holstein model (as well as the Hubbard and extended Hubbard models) at 10% hole doping using a 120-site chain to observe relatively long-distance behavior. We measure equal-time correlation functions of the form $\langle \hat{O}_{i+r} \hat{O}_i \rangle$, averaged over 5 reference points (i.e., $i = L/4 - 1, L/4, \ldots, L/4 + 3$) for each $r$, where $r$ is the distance between two sites along the chain between 0 to $L/2$. In this way, the measurements fall roughly within the center half of the chain to reduce boundary effects.

Our results suggest that the ground state of the Hubbard-extended Holstein model in 1D is consistent with a Luttinger liquid (LL)[35], as evidenced by the slow decay of the single-particle Green's function defined as $G_\sigma(r) = \langle \hat{c}^\dagger_{i+r,\sigma} \hat{c}_{i,\sigma} \rangle$. Specifically, $G_\sigma(r)$ as shown in Fig. 5a can be very well fitted by a power law, i.e., $G_\sigma(r) \sim r^{-K_G}$. The decaying behavior of the single-particle Green function for each of the three different models is qualitatively consistent with the Luttinger exponent $K_G \sim 1$. We provide the value of $K_G$ extracted from each model in Table 1. For completeness, we have calculated the spin-spin correlation function defined as $F(r) = \langle \mathbf{S}_{i+r} \cdot \mathbf{S}_i \rangle$. As shown in Fig. 5b, $F(r)$ also appears to decay as a power law, $F(r) \sim r^{-K_s}$, but with a larger exponent than the single-particle correlation, $K_s > K_G$, also consistent LL behavior. Note that the extended $e$–$ph$ interaction produces a larger suppression of the spin-spin correlations, resulting in the largest $K_s$ among the three models. The charge density-density fluctuation correlations (see Fig. 5c), defined as $D(r) = \langle \hat{n}_{i+r} \hat{n}_i \rangle - \langle \hat{n}_{i+r} \rangle \langle \hat{n}_i \rangle$, also appear quasi-long-ranged with a Luttinger exponent $K_c$, also shown in Table 1.

The most intriguing aspect of the extended interactions may be their influence on superconductivity, tested through the equal-time spin-singlet superconducting pair-field correlation function, $P(r) = \langle \hat{\Delta}^\dagger_{i+r} \hat{\Delta}_i \rangle$, where $\Delta_i = \frac{1}{\sqrt{2}}(\hat{c}_{i\uparrow} \hat{c}_{i+1,\downarrow} - \hat{c}_{i\downarrow} \hat{c}_{i+1,\uparrow})$ is the spin-singlet pair-field annihilation operator. As expected for a LL, $P(r)$ decays as a power law, with $K_{sc} > 2$ for all three models, as shown in Fig. 5d and Table 1. Most importantly, not only does the nearest-neighbor attractive interaction enhance $P(r)$, but the extended $e$–$ph$ coupling also produces a noticeably smaller $K_{sc}$ compared to the Hubbard model alone. Taken together, while the extended $e$–$ph$ interaction itself does not qualitatively alter the ground state of the system in 1D, it does quantitatively enhance the strength of singlet superconducting pair-field correlations and suppress spin-spin correlations.

## Discussion

In summary, the inclusion of extended electron-lattice couplings is crucially important for reproducing many of the observed spectral features in ARPES. The extended $e$–$ph$ coupling reproduces well the intensity and doping dependence of the hf-feature, and the reduced $3k_F$ feature and gives the right doping dependence of the holon band width. As more experimental results emerge for doped 1D systems, it

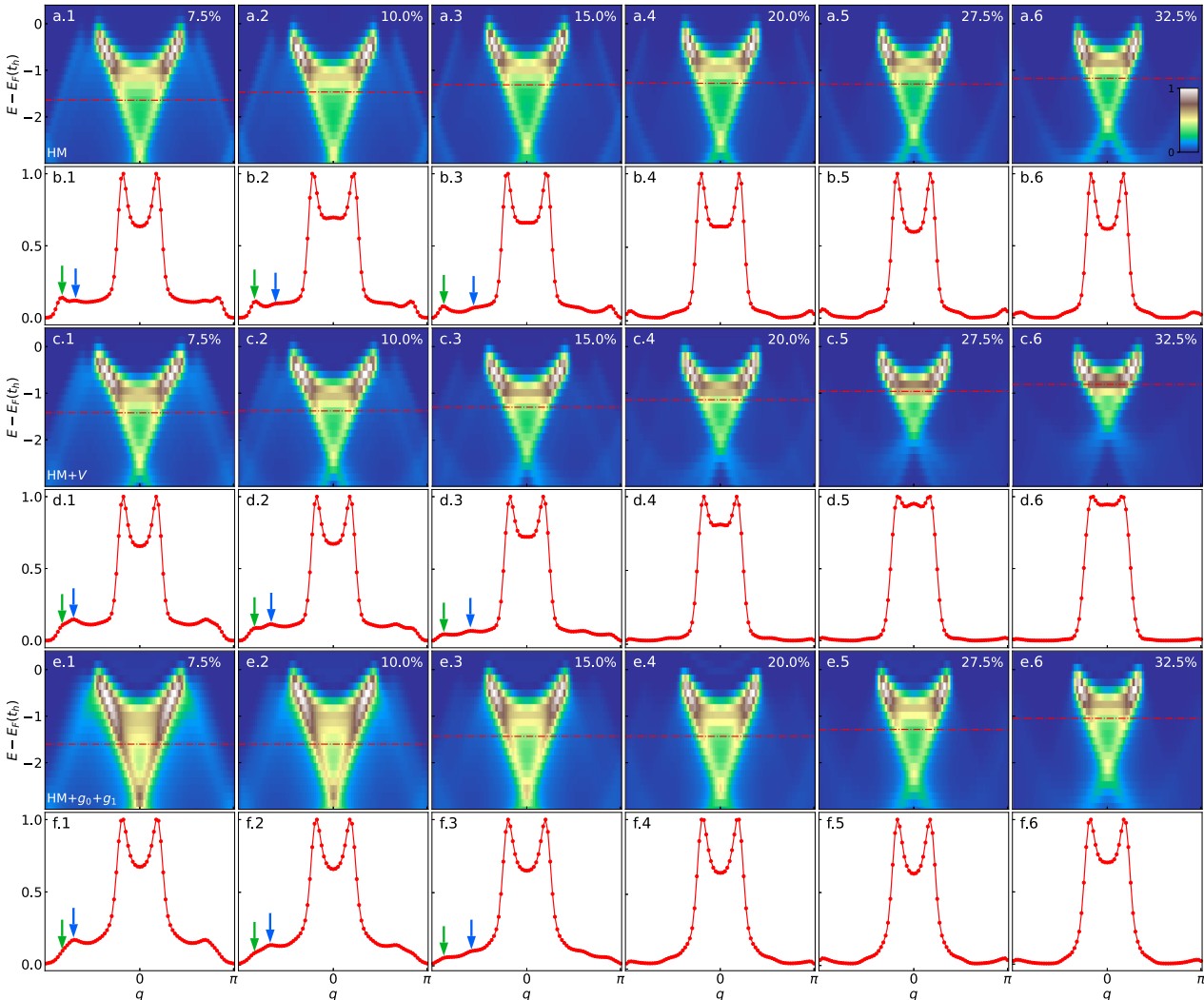

**Fig. 3 | Single-particle spectra for different models. a** The Hubbard model (HM), **c** the extended Hubbard model (HM + $V$), and **e** the Hubbard-extended Holstein model (HM + $g_0$ + $g_1$), with increasing doping from columns 1–6. Panels **b**, **d**, **f** show representative momentum distribution curves (MDCs), corresponding to the cuts given by the red dashed line for each of the spectra in (**a**, **c**, **e**), respectively. The MDCs are chosen -$t_h$ above the bottom of the holon branch to ensure that the main holon peaks are at roughly the same position for different dopings and for different models, providing equivalent MDCs for comparison. The green and blue arrows mark the positions of the $3k_F$ and hf branches, respectively. One can clearly see that at lower doping (<20%), adding nearest neighbor attraction $V$ or extended $e-ph$ coupling can enhance the hf branch while suppressing the $3k_F$ branch. Above 20% doping, both peaks fade away quickly. Here, the energy and momentum broadening of the spectra are $\sigma_\omega = 0.18t_h$ and $\sigma_k = 2\pi/L$. The lesser Green's functions data corresponding to all the single-particle spectral functions displayed here can be found in Fig. S5 in Supplementary Information.

would be beneficial to check the impact of $e-ph$ coupling on other measurements, such as the dynamical spin structure factor and phonon dispersion.

Our results show that the ground state of the 1D Hubbard-extended Holstein model remains a Luttinger liquid with a single-particle correlation exponent $K_G$-1 and subdominant superconducting correlations. However, quantitatively, the extended $e-ph$ coupling helps to suppress the spin correlations $F(r)$ while simultaneously enhancing the superconducting pair-field correlations $P(r)$. Importantly, while the inclusion of a simple, effective nearest-neighbor attractive interaction to approximate the extended $e-ph$ coupling can produce a similar enhancement of the hf-branch, it fails to produce the right holon band width. It also enhances $P(r)$ and gives a $K_{sc}$ close to the one produced by the extended $e-ph$ coupling but overestimates the magnitude of $P(r)$ and does not suppress $F(r)$ as effectively as the extended $e-ph$ coupling.

It is of course an open and interesting question to determine whether the agreement between numerical results for the Hubbard-

extended Holstein model and ARPES translates to dimensions greater than 1. In 2D $t'$ will play an important role, as has been shown in previous DMRG calculations on cylinders[7–13] where $t'$ can help to stabilize quasi-long-range superconducting correlations. However, these superconducting correlations are subdominant to charge density wave (CDW) correlations and become weaker on wider cylinders. A recent DMRG study on 4-leg ladders has shown that an effective nearest-neighbor electron–electron attraction can result in dominant quasi-long-range $d$-wave superconducting correlations on the hole-doped side with negative $t'$, where the crossover between dominant super-conducting and CDW correlations occurs near $V \sim -t_h$[36]. Yet that ground state remains qualitatively consistent with a Luther-Emery liquid, as found in the simple Hubbard model with $t'$ on the same ladder. As it appears that power-law decay of superconducting corre-lations cede to a short-range exponential decay of correlations as the hole-doped ladder system goes to 2D, a boost of superconducting pairing from extended electron–lattice coupling could be pivotal to both qualitatively and quantitatively change the nature of the ground

state. While this remains a topic of investigation, our results encourage additional study on the influence of phonon degrees of freedom in 2D models, which finally may help to realize a *d*-wave superconducting ground state.

## Methods

### Time-evolving block decimation

We use the time-evolving block decimation (TEBD) scheme, which was introduced by Vidal[25] and later incorporated into the DMRG algorithm by White[26] for time evolution. TEBD utilizes a Trotter-Suzuki decomposition of the time evolution operators; and we consider the second-order TEBD (TEBD2) scheme, which uses the decomposition

$$e^{-iH\delta t} = \prod_{i=0}^{L-2} e^{-ih_{i,i+1}\delta t/2} \prod_{i=L-2}^{0} e^{-ih_{i,i+1}\delta t/2} + O(\delta t^3). \qquad (4)$$

Here, we consider a Hamiltonian that contains only nearest-neighbor couplings, and $h_{i,i+1}$ contains terms involving only sites $i$ and $i+1$ along a 1D chain. The time evolution operator can be applied to the wave function during a DMRG sweep, replacing the ground state solving step by applying the gate $e^{-ih_{i,i+1}\delta t/2}$ when sites $i$ and $i+1$ are at the center. In this way, we avoid numerical errors due to truncation when site $i$ or $i+1$ is in the system or environment block. Specifically, we can apply gates $e^{-ih_{0,1}\delta t/2}$, $e^{-ih_{1,2}\delta t/2}$, ..., $e^{-ih_{L-2,L-1}\delta t/2}$ in the left-to-right sweep, then reverse sweep direction, and apply all the reverse gates in

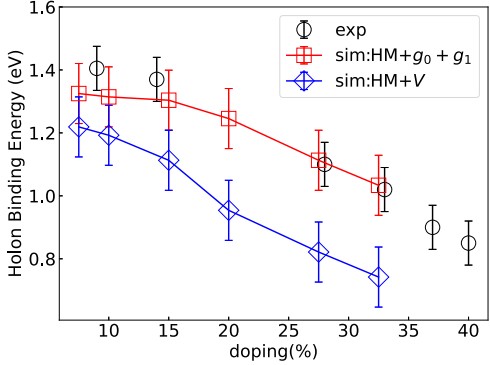

**Fig. 4 | Comparison of experimental and simulated holon binding energy at momentum $k = 0$.** The experiment data (open circle) are taken from ref. [19]. We use the broadening as the error bar for the simulated data. For the holon binding energy, the Hubbard-extended Holstein model (open square) matches the experiment data very well, while the extended-Hubbard model (open diamond) deviates from the experiment at higher doping. Here we take $t_h = 530$ meV.

the right-to-left sweep. Thus all gates for a one-time step can be applied by a complete left-to-right and right-to-left sweep[26,27].

### Local basis optimization

The unbounded phonon Hilbert space on each site presents a challenge for wave function-based numerical techniques. It is usually inefficient to naively truncate the Hilbert space, keeping only the first $N$ bare phonon basis on each site ($|0\rangle, |1\rangle, ..., |N-1\rangle$), especially when the $e$–$ph$ coupling is strong, and many bare phonons are needed for convergence. This becomes prohibitive for techniques like exact diagonalization (ED) and also may make DMRG simulations difficult, if not unfeasible. One method to solve this problem is to perform an LBO, truncating the local phonon Hilbert space to a few optimal basis[28], similar to truncation of the system and environment Hilbert space blocks in traditional DMRG ground state calculations. This approach works very well, often with only 2 to 3 optimal phonon basis elements that can provide good ground state convergence in the Holstein model[28].

LBO has been extended for time evolution, a dynamical LBO, where the phonon basis on each site is optimized in a position- and time-dependent manner[29]. Figure 1b illustrates how to perform dynamical LBO, where the local Hilbert space of dimension $D$ is optimally truncated to $d \ll D$. During time evolution, We first enlarge the Hilbert space of each of the two center sites to dimension $D$, then apply the Trotter gate in this enlarged Hilbert space to reduce errors due to truncation. Finally, we truncate the local Hilbert space back to dimension $d$, which significantly reduces the numerical cost for truncating the system or environment block[29]. In our calculations, both ground state LBO and dynamical LBO for time evolution provide reasonable convergence for the Hubbard-extended Holstein model.

### Lesser Green's function and Fourier transform

Using time evolution, we can calculate the lesser Green's function $\mathcal{G}^<_{mn\sigma}(t) = i\langle \hat{c}^\dagger_{m\sigma}(t)\hat{c}_{n\sigma}(0)\rangle$. To do so, we need to time evolve both the ground state $|G(t)\rangle = e^{-iHt}|G\rangle$ and the removal state $|R_{n\sigma}(t)\rangle = e^{-iHt}\hat{c}_{n\sigma}|G\rangle$, such that $\mathcal{G}^<_{mn\sigma}(t) = i\langle G(t)|\hat{c}^\dagger_{m\sigma}|R_{n\sigma}(t)\rangle$. The single-particle removal spectra, which can be compared to ARPES spectra, are obtained by a Fourier transform of the lesser Green's function

$$\mathcal{A}^-(k,\omega) = \int_{-\infty}^{\infty} \frac{dt\, e^{i\omega t}}{2\pi i} \sum_{mn\sigma} \frac{e^{-ik(r_n - r_m)}}{L^2} \mathcal{G}^<_{mn\sigma}(t), \qquad (5)$$

where $k$ represents momentum along the chain, $\omega$ is frequency, and $L$ is the length of the chain. One typically fixes the position index $n$ to the center of the chain ($n = L/2 - 1$ or $n = L/2$); correspondingly, the summation runs only over index $m$ with a normalization factor $1/L$ rather than $1/L^2$. To ensure reflection symmetry for chains with an even number of sites, we average over the spectra obtained from $\mathcal{G}^<_{m,L/2-1,\sigma}$

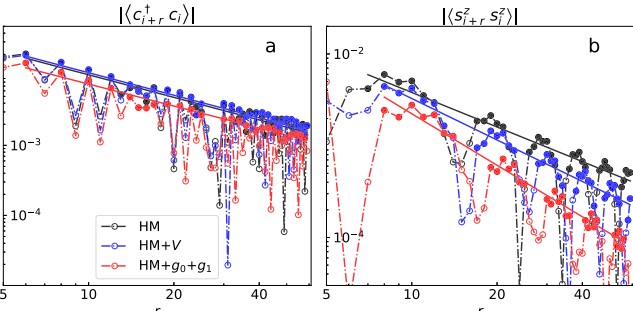
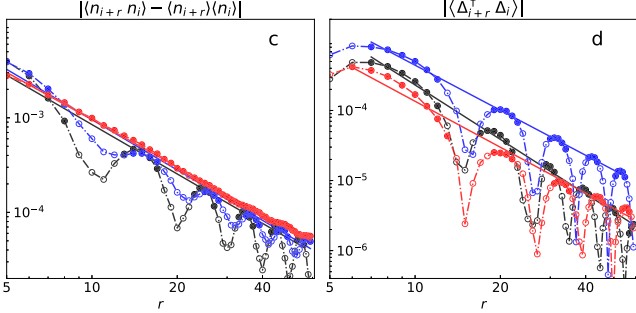

**Fig. 5 | Correlation functions for different models. a** Single-particle Green's function. **b** Spin-spin correlation. **c** Charge density–density fluctuation correlation. **d** Spin-singlet superconducting pair-field correlation. Correlations for the Hubbard (black) and extended Hubbard (blue) models are plotted for comparison. The

Hubbard-extended Holstein model (red) results were obtained for $\omega_0 = 0.2t_h$, $g_0 = 0.3t_h$, and $g_1 = 0.15t_h$. The straight line fits follow a power law decay $-r^{-K}$ to extract effective Luttinger exponents for different correlation functions. Filled circles show data used for fitting[9].

**Table 1 | Comparison of exponents extracted for various correlation functions for different models**

| Model | $K_G$ | $K_s$ | $K_c$ | $K_{sc}$ |
|---|---|---|---|---|
| HM | 1.07(6) | 1.25(5) | 1.74(5) | 2.58(4) |
| HM + V | 1.07(5) | 1.50(7) | 1.77(4) | 2.14(3) |
| HM + $g_0$ + $g_1$ | 1.04(4) | 1.87(6) | 1.66(1) | 2.18(6) |

The exponents $K$ are extracted from the fits $\sim r^{-K}$ shown in Fig. 5.

and $\mathcal{G}^<_{m,L/2,\sigma}$. We evolve the system from time 0 to time $T$ before the excitation propagates to the boundary of the chain. To regularize the Fourier transform due to the finite cutoff time, we use a window function $\mathcal{W}(t)$, where $\mathcal{W}(T) \sim 0$, which broadens the spectra and acts as a frequency resolution convolution $\mathcal{A}(k,\omega) = \mathcal{A}_\_(k,\omega) * \mathcal{F}[\mathcal{W}(t)]$. We use a Gaussian window function with frequency domain standard deviation $\sigma_\omega$. We also convolve the spectra in momentum using a Gaussian filter with a standard deviation of one momentum spacing $\sigma_k = 2\pi/L$, which removes high-frequency noise and smooths the spectra.

#### Convergence
We keep $m = 800$ states during time evolution which produces a truncation error below $1 \times 10^{-6}$ for the Hubbard and extended Hubbard models and below $1 \times 10^{-5}$ for the Hubbard-extended Holstein model. The time step is fixed at $\delta t = 0.025 t_h^{-1}$, and we evolve the system up to $T = 20 t_h^{-1}$. For dynamical LBO, we keep 20 bare phonon basis ($D = 80$ in Fig. 1b) and truncate to a basis of 3 optimal phonons ($d = 12$ in Fig. 1b) on every site. This results in a phonon truncation error below $1 \times 10^{-4}$.

Time evolution convergence with respect to both $\delta t$ and $m$ (up to 1200) has been checked on the 80-site chain for the Hubbard model. Adding phonons makes the calculations quiet expensive and convergence with respect to the local bare basis dimension $D$ and optimal basis dimension $d$ have been checked on an 8-site chain, where many more bare phonons can be kept and time evolution without dynamical LBO can be carried out for benchmark. For $D = 80$ and $d = 12$, both the ground state energy and the lesser Green's function (see Figs. S1–S4 in Supplementary Information) converge well on the short chain for the Hubbard-extended Holstein model, and time evolution on the 80-site chain can be completed for a reasonable computational cost. We are also able to use $m = 900$, $D = 100$, and $d = 12$ on the 80-site chain, which gives the same results with $m = 800$, $D = 80$, and $d = 12$.

In the ground state correlation functions calculation, we keep up to $m = 1000$ states, which results in a truncation error ranging from $6 \times 10^{-9}$ to $3 \times 10^{-7}$, depending on the model and $e$–$ph$ coupling strength. We truncate a bare phonon basis of up to 40 ($D = 160$) to an optimal phonon basis of up to 4 ($d = 16$). This results in a phonon basis truncation error ranging from $2 \times 10^{-8}$ to $3 \times 10^{-7}$, depending on the $e$–$ph$ coupling strength.

## Data availability
All data that support the findings of this study are present in the paper and the Supplementary Information. Additional data related to the study are available from the corresponding author upon reasonable request.

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

## Acknowledgements

The authors would like to thank Hongchen Jiang, Yao Wang and Zhuoyu Chen for helpful discussions and suggestions. This work was supported by the U.S. Department of Energy, Office of Basic Energy Sciences, Division of Materials Sciences and Engineering, under Contract No. DE-AC02-76SF00515. The computational results utilized the resources of the National Energy Research Scientific Computing Center (NERSC) supported by the U.S. Department of Energy, Office of Science, under Contract No. DE-AC02-05CH11231. Some of the computing for this project was performed on the Sherlock cluster. We would like to thank Stanford University and the Stanford Research Computing Center for providing computational resources and support that contributed to these research results.

## Author contributions

T.P.D. and Z-X.S. designed the project. T.T. implemented the tDMRG algorithm, carried out the simulation, analyzed the data and prepared the first draft of the manuscript. B.M. and C.P. contributed to data analysis. All authors contributed to the discussion and the writing of the paper.

## Competing interests

The authors declare no competing interests.
