## [Peer Review File · Nature Communications]

REVIEWER COMMENTS

Reviewer #1 (Remarks to the Author):

The present work is motivated by recent photo-emission experiments on 1D cuprate chains which could theoretically be explained by an extended Hubbard model including a nearest-neighbor attraction. This attraction has been argued to emerge from long-range electron-phonon coupling in a subsequent paper by an approximate method by a group including some of the authors of the present work. Now, the present work contains accurate unbiased density matrix renormalization group and dynamical calculations of an Hubbard-extended Holstein model, i.e. a Hubbard model whose electrons are coupled to onsite and nearest-neighbor phonons. In this way, the authors do not rely on an effective nearest-neighbor attraction to explain experimental data, but results directly follow from the electron-phonon coupling. The two main results of the paper are:

1) calculation of the single-particle spectral function: at low doping the e-ph coupling causes both the suppression of the $3k_F$ branch and the enhancement of the holon folding branch consistent with experimental data. these features disappear with stronger doping also in agreement with the experiments. In addition, the holon binding energy at $k=0$ is more consistent with experimental data in the e-ph model than in the Hubbard model with effective attraction.

2) ground state wave function: the ground state at 10% doping is a Luttinger liquid for all three models (pure Hubbard, Hubbard+nearest neighbor, Hubbard plus phonons). However, in the e-ph model, the singlet pair-pair correlations are enhanced which could be an indication for enhancement of superconductivity in an analogous 2D model.

I think the work represents a significant contribution to the understanding of the nature of the electronic states in the 1D cuprates and the role of e-ph interactions. The work weakens possible concerns as to whether phonons can account for the spectral features in the experiments, as previous explanations went over the detour of an effective nearest-neighbor attraction in the Hubbard model. In contrast, it shows that directly solving the model with e-ph coupling reproduces the experimental results even better. The methods are unbiased many-body quantum techniques and the results are analyzed carefully, especially regarding the local basis optimization to effectively cut the number of phononic degrees of freedom. The ultimate goal is of course to understand the role of phonons in the 2D cuprates and their connection to superconductivity, a topic of immense interest to the entire condensed matter community and beyond. The ground state data supports that such e-ph coupling can enhance superconducting pairing tendencies, and although this is speculative, it could be an indication for the effects of phonons in 2D. The authors comment on that direction, but do not over-interpret their results. To recapitulate, I believe the the manuscript is

clear and well-written, the methods appropriate and the results are significant and appeal to a wide readership.

I have some specific comments that the authors should address:

- The authors state that the time evolution data has converged in both bond dimension m and time step dt . I believe that this is true for the observables used in the main text, but does it also hold for the entanglement entropy in Fig. E5 or is this flattening a finite- m effect?

- Caption of Fig. 1: "Each site i has d optimized basis" is not very clear, maybe "Each site i has an optimized basis of dimension d "?

Reviewer #2 (Remarks to the Author):

The authors studied the properties of one-dimensional cuprates. They conducted time-dependent density matrix renormalization group (tDMRG) calculation on the Hubbard model, extended-Hubbard model and Hubbard-extended Holstein model, and found that the results from Hubbard-extended Holstein model are most consistent with the ARPES data. By analyzing the Luttinger liquid exponents, they found that in the Hubbard-extended Holstein model, the electron-phonon coupling enhances the superconducting pairing while suppressing the spin-spin correlation, though the ground state remains a Tomonaga-Luttinger liquid, as expected for one dimension (1-d). The results are novel, solid and of significance to the community. The paper is clearly written. I recommend it for publication in Nature communications with two suggestions:

1. In the Hubbard-type models for 1-d, the next nearest neighbor hopping is not considered, most likely because the lattice distance is $2a$. With the hope that the conclusion may shed light on 2-d cuprates, the next nearest neighbor hopping t' would play an important role since the lattice distance is $2^{0.5} a$. The authors may want to briefly discuss how t' may modify the current picture if extending to 2-d.

2. The author shall include color bars for Fig.3.

Response to the Report of Reviewer #1

Reviewer #1 (Remarks to the Author):

The present work is motivated by recent photo-emission experiments on 1D cuprate chains which could theoretically be explained by an extended Hubbard model including a nearest-neighbor attraction. This attraction has been argued to emerge from long-range electron-phonon coupling in a subsequent paper by an approximate method by a group including some of the authors of the present work. Now, the present work contains accurate unbiased density matrix renormalization group and dynamical calculations of an Hubbard-extended Holstein model, i.e. a Hubbard model whose electrons are coupled to onsite and nearest-neighbor phonons. In this way, the authors do not rely on an effective nearest-neighbor attraction to explain experimental data, but results directly follow from the electron-phonon coupling. The two main results of the paper are:

1) calculation of the single-particle spectral function: at low doping the e-ph coupling causes both the suppression of the $3k_F$ branch and the enhancement of the holon folding branch consistent with experimental data. these features disappear with stronger doping also in agreement with the experiments. In addition, the holon binding energy at $k=0$ is more consistent with experimental data in the e-ph model than in the Hubbard model with effective attraction.

2) ground state wave function: the ground state at 10% doping is a Luttinger liquid for all three models (pure Hubbard, Hubbard+nearest neighbor, Hubbard plus phonons). However, in the e-ph model, the singlet pair-pair correlations are enhanced which could be an indication for enhancement of superconductivity in an analogous 2D model.

I think the work represents a significant contribution to the understanding of the nature of the electronic states in the 1D cuprates and the role of e-ph interactions. The work weakens possible concerns as to whether phonons can account for the spectral features in the experiments, as previous explanations went over the detour of an effective nearest-neighbor attraction in the Hubbard model. In contrast, it shows that directly solving the model with e-ph coupling reproduces the experimental results even better. The methods are unbiased many-body quantum techniques and the results are analyzed carefully, especially regarding the local basis optimization to effectively cut the number of phononic degrees of freedom. The ultimate goal is of course to understand the role of phonons in the 2D cuprates and their connection to superconductivity, a topic of immense interest to the entire condensed matter community and beyond. The ground state data supports that such e-ph coupling can enhance superconducting pairing

tendencies, and although this is speculative, it could be an indication for the effects of phonons in 2D. The authors comment on that direction, but do not over-interpret their results. To recapitulate, I believe the the manuscript is clear and well-written, the methods appropriate and the results are significant and appeal to a wide readership.

We thank the reviewer for positive assessment and feedback on our work, as well as a detailed and thoughtful summary of the key points of the manuscript.

I have some specific comments that the authors should address:

- The authors state that the time evolution data has converged in both bond dimension m and time step dt . I believe that this is true for the observables used in the main text, but does it also hold for the entanglement entropy in Fig. E5 or is this flattening a finite- m effect?

We thank the reviewer for asking this question. We agree with the reviewer that the entropy can appear smaller than the true value if the bond dimension m is too small. As the entropy grows in later time, ideally one should use a larger m . To fully answer the reviewer's question, a larger m should be used, which can be checked easily for the HM and HM+V models, where $m=800$ entropy values only differ slightly from the $m=1200$ results at late times. The relative error in entropy is around 1-2% for the last time step. For HM+ g_0+g_1 , we were only able to check $m=900$ and $D=100$ due to the large numerical cost associated with phonons. Entropy values for $m=900$ and $D=100$ overlap with those for $m=800$ and $D=80$. Based on these results, we believe the entropy values on the 80-site chain are at the very least near convergence, such that the flattening is not an artifact due to finite m . We also note that small errors in later time have a very limited impact on $A(k,w)$ due to the use of a Gaussian window function before applying the Fourier transformation. This assigns diminishing weights to data at later times. We have added a clarification about the entropy convergence in the caption to Fig.E5.

- Caption of Fig. 1: "Each site i has d optimized basis" is not very clear, maybe "Each site i has an optimized basis of dimension d "?

We thank the reviewer for the suggestion. We have changed the wording in the caption of Fig.1.

Response to the Report of Reviewer #2

Reviewer #2 (Remarks to the Author):

The authors studied the properties of one-dimensional cuprates. They conducted time-dependent density matrix renormalization group (tDMRG) calculation on the Hubbard model, extended-Hubbard model and Hubbard-extended Holstein model, and found that the results from Hubbard-extended Holstein model are most consistent with the ARPES data. By analyzing the Luttinger liquid exponents, they found that in the Hubbard-extended Holstein model, the electron-phonon coupling enhances the superconducting pairing while suppressing the spin-spin correlation, though the ground state remains a Tomonaga–Luttinger liquid, as expected for one dimension (1-d). The results are novel, solid and of significance to the community. The paper is clearly written. I recommend it for publication in Nature communications with two suggestions:

We would like to thank the reviewer for their positive assessment and thoughtful comments on our manuscript.

1. In the Hubbard-type models for 1-d, the next nearest neighbor hopping is not considered, most likely because the lattice distance is $2a$. With the hope that the conclusion may shed light on 2-d cuprates, the next nearest neighbor hopping t' would play an important role since the lattice distance is $2^{0.5} a$. The authors may want to briefly discuss how t' may modify the current picture if extending to 2-d.

Yes, we agree that t' will play an important role in 2-d cuprates, as has been shown in previous DMRG calculations on cylinders (Ref.[7-13]) where t' can help to stabilize quasi-long-range superconducting correlations on the hole-doped side, which is subdominant to CDW order and actually becomes weaker on wider cylinders. The discrepancy on the hole-doped side to experiment in cuprates may indicate additional ingredients are needed. However, we note that the phase diagram differs between the Hubbard model on the 1D chain and cylinders, so a direct extension of the 1D conclusion to 2D with both t' and electron-phonon coupling is probably not very appropriate. But the paper by some of us on the four-leg cylinder (Ref.[36]) studies the hole-doped single-band Hubbard model with near-neighbor attractive V and negative next-neighbor t' , where the phase qualitatively continues to be the same as without the attractive V , but SC is enhanced, eventually becoming dominant when compared to CDW if V is sufficiently strong. This indicates that an extended electron-phonon coupling may help to boost superconductivity on the hole-doped side of the Hubbard model with appropriate t' , and may provide important physics bringing the microscopics

closer to that of the hole-doped cuprates. We thank the reviewer for pointing out this and have added a brief comment on this in the discussion.

2. The author shall include color bars for Fig.3.

We thank the reviewer for pointing this out. For completeness, we have added color bars to Fig.3.

Summary of Modifications

All the following modifications are colored in red in the manuscript.

- (1) In Fig. 1's caption, we changed "Each site i has d optimized basis" to "Each site i has an optimized basis of dimension d ".
- (2) In the Method/convergence section, we added comparison to larger m and D .
- (3) In the caption of Fig. E5, we add clarification about the convergence of the entropy.
- (4) We added a brief comment about the influence of electron-phonon coupling and t' in 2D in the discussion.
- (5) We added color bars to Fig. 3.

We also made the following modifications (not colored in red):

- (1) Added section titles such as Introduction, Result and Discussion.
- (2) Added titles for each figure in bold font.
- (3) Added a Data Availability statement after the methods section.
- (4) Added an Author Contribution statement.
- (5) Added a Competing Interests statement.

REVIEWERS' COMMENTS

Reviewer #1 (Remarks to the Author):

The authors have responded satisfactorily to all the referee requests. I therefore recommend publication in the revised version.

Reviewer #2 (Remarks to the Author):

My concerns are addressed satisfactorily.